

# Assessing the effects of time-dependent restrictions and control actions to flatten the curve of COVID-19 in Kazakhstan

Ton Duc Do[1,*], Meei Mei Gui[2] and Kok Yew Ng[3,4,*]

[1] Department of Robotics and Mechatronics, School of Engineering and Digital Sciences, Nazarbayev University, Nur-Sultan, Kazakhstan
[2] School of Chemistry and Chemical Engineering, Queen's University Belfast, Belfast, United Kingdom
[3] Engineering Research Institute, University of Ulster, Belfast, United Kingdom
[4] Electrical and Computer Systems Engineering, School of Engineering, Monash University Malaysia, Bandar Sunway, Malaysia
* These authors contributed equally to this work.

## ABSTRACT

This article presents the assessment of time-dependent national-level restrictions and control actions and their effects in fighting the COVID-19 pandemic. By analysing the transmission dynamics during the first wave of COVID-19 in the country, the effectiveness of the various levels of control actions taken to flatten the curve can be better quantified and understood. This in turn can help the relevant authorities to better plan for and control the subsequent waves of the pandemic. To achieve this, a deterministic population model for the pandemic is firstly developed to take into consideration the time-dependent characteristics of the model parameters, especially on the ever-evolving value of the reproduction number, which is one of the critical measures used to describe the transmission dynamics of this pandemic. The reproduction number alongside other key parameters of the model can then be estimated by fitting the model to real-world data using numerical optimisation techniques or by inducing ad-hoc control actions as recorded in the news platforms. In this article, the model is verified using a case study based on the data from the first wave of COVID-19 in the Republic of Kazakhstan. The model is fitted to provide estimates for two settings in simulations; time-invariant and time-varying (with bounded constraints) parameters. Finally, some forecasts are made using four scenarios with time-dependent control measures so as to determine which would reflect on the actual situations better.

## INTRODUCTION

According to the World Health Organization (WHO), more than 41.5 million diagnosed cases related to COVID-19 with almost 1.15 million deaths have been reported globally as of October 23, 2020 (*World Health Organization, 2020*). Although some countries such as South Korea (*Exemplars in Global Health, 2020*), Japan (*The Nippon Communications*

Corresponding author
Kok Yew Ng, mark.ng@ulster.ac.uk

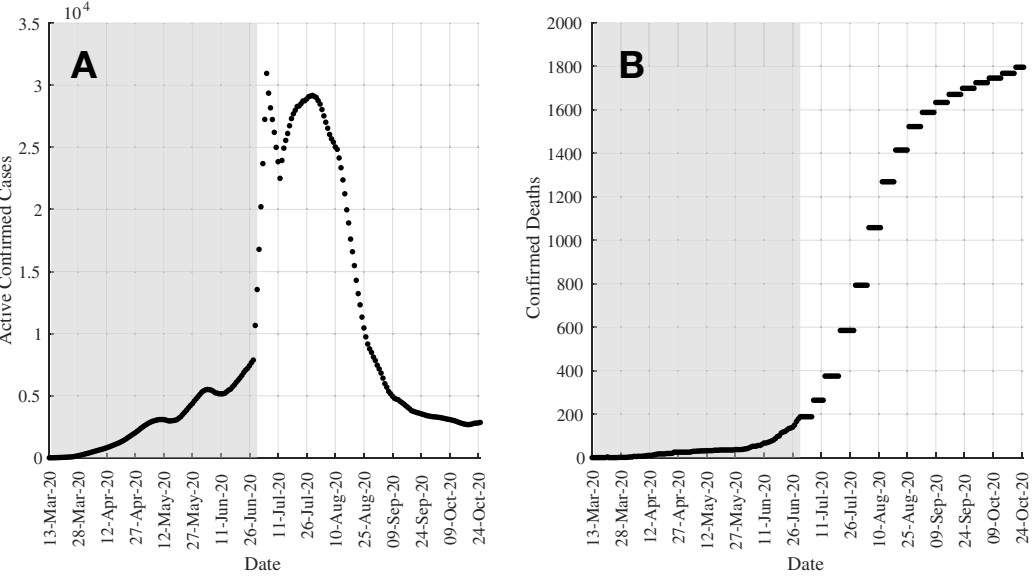

**Figure 1 Plots showing the up-to-date cases in Kazakhstan as of October 25, 2020.** (A) shows the 7-day moving average of active confirmed cases whilst (B) shows the confirmed deaths. Data in both graphs are plotted since the first cases were reported on March 13, 2020. The shaded regions show the range of data (up till Day 108 (June 28, 2020)) used to understand the effectiveness of control actions in this study.

*Foundation, 2020*), New Zealand (*Klein, 2020*; *The Japan Times, 2020*), Malaysia (*The Edge Markets, 2020*), and Vietnam (*The Japan Times, 2020*) had this pandemic under control during its early stages, many countries are now battling the second (*Cacciapaglia, Cot & Sannino, 2020*) or even the third wave of the pandemic (*Popescu, 2020*). In Kazakhstan, the first cases were reported on March 13, 2020 (*Kazinform International News Agency, 2020*), which was quite late compared to other countries within the region. Right after that, the Kazakhstan government implemented aggressive intervention methods such as lockdowns of its main cities, social distancing, quarantines, and closure of schools. Despite those efforts, the spread of COVID-19 was still developing in the country before it was eventually brought under control as seen in Fig. 1, which shows the 7-day moving average of active confirmed cases (left subfigure) and the confirmed deaths (right subfigure) in Kazakhstan.

As such, mathematical models are essential to help analyse the dynamics of the spread of COVID-19. One of the conventional mathematical models, namely the deterministic compartmental SIR (susceptible, infectious, and recovered/removed) model, has been used to predict viral or bacterial transmission diseases such as severe acute respiratory syndrome (SARS), tuberculosis, meningitis, cholera, measles, influenza A (H1N1), and HIV (*Brauer, Castillo-Chavez & Castillo-Chavez, 2012*; *Rock et al., 2014*). The SIR model demonstrates the transportation of individuals as they go through three mutually exclusive stages (compartments) of infection during the epidemic: susceptible (S), infected (I), and recovered/removed (R), where the disease transmission rates with respect to time can then be simulated. The SIR model and its variations have also been used to model the COVID-19 pandemic. For example, a discrete-time SIR model was reported in

*Anastassopoulou et al. (2020)*, whilst a control-oriented SIR model was presented in *Casella (2021)*. Also, in *Wu et al. (2020)*, the SIR model was used to estimate the clinical severity of COVID-19.

A commonly-used variation of the SIR model is the SEIR model, where an exposed (E) compartment is added to model a subpopulation of people who have been exposed to the disease but have yet to become infectious (I) (*Lin et al., 2020*). This seems to be a more suitable model to describe the dynamics of COVID-19 as it has been established that there exists an incubation period where an exposed person is pre-symptomatic before starting to show symptoms and become infectious (*Lauer et al., 2020*). In the literature, there are some studies on using the SEIR model and its variants to model the COVID-19 outbreak. For example, *Keeling et al. (2020)* extended the SEIR model to include age-specific studies to provide short-term forecasts and to analyse the impact of compliancy (or the lack of) onto the transmission dynamics of COVID-19 in the UK. Meanwhile, *Bae, Kwon & Kim (2020)* and *Piccolomini & Zama (2020)* used the SEIRD model to model the spread of COVID-19 in Korea and Italy, respectively. *Piccolomini & Zama (2020)* also used time-dependent parameters in their model. However, these works did not provide the analysis on the effects of time-dependent control actions and restrictions taken by the respective countries to flatten the curve of COVID-19.

One of the main challenges to predict the evolution of the pandemic is the curve-fitting problem. The Levenberg–Marquardt (LM) and trust-region-reflective (TRR) algorithms are amongst two of the solutions that can be used to help solve this problem (*Moré, 1978*; *Sorensen, 1982*). They were first introduced in the 1960s to solve nonlinear least squares problems. The least squares problems address the issues of fitting a parameterised function to a set of measured data points by minimising the sum of the squares of the errors between the reference data and the prediction from the model. This could be used to solve the parameter estimation problem for compartmental models such as the SIR and SEIR models. Basically, the LM algorithm is the combination of the gradient descent method and Gauss–Newton method (*Haddout & Rhazi, 2015*). The LM method acts more like a gradient-descent method when the parameters are far from their optimal value, and becomes more like the Gauss–Newton one when the parameters are close to the optimal value. However, the LM algorithm may not converge nicely if the initial guess is too far from the optimum, which can be prevented by using the TRR algorithm.

The article presents the assessment and forecast of the COVID-19 pandemic using a modified SEIRD model to estimate the time-invariant and time-varying (with bounded constraints) parameters of the model, and also to analyse the effects of time-dependent control actions onto the kinetics of the spread of the virus. The SEIRD (susceptible, exposed, infectious, recovered, and death) model is a variant of the SEIR model, where the death (D) compartment is used to represent the fraction of the infectious subpopulation who have unfortunately succumbed to the disease. As of October 25, 2020, there have been more than 110,400 confirmed cases and 1,796 deaths recorded in Kazakhstan. The first wave of the COVID-19 outbreak in Kazakhstan is studied in detail as a case study. This is both critical and necessary in order to inform on the response of the transmission dynamics of COVID-19 to the control actions taken by the authorities in the country.

The data used for this study is represented by the shaded regions (data up till Day 108 (June 28, 2020)) in the plots in Fig. 1. First, the initial fitting based on the real data in Kazakhstan is performed using TRR algorithm using both constant and bounded time-related parameters to obtain crucial information such as the reproduction number of the pandemic. Further simulations are also carried out by inducing ad-hoc control actions into the model. These results are able to help translate the transmission dynamics of COVID-19 as well as the effects and efficacy of the control actions taken in the country. Then some predictions are made based on these estimated parameters, where four scenarios of reinstating of intervention measures are introduced at different times. Simulation results show that one of the scenarios is able to describe the current COVID-19 situation in Kazakhstan, and hence can be used to further inform on the future plan in controlling the pandemic, especially during the unfortunate event of a second wave. Therefore, this article will focus mainly on the solution of the inverse problem of the model using TRR as well as other methods that will be discussed later in this article. This article does not provide the solution of the forward problem of the model. Despite the discussion on the predictions made by the proposed model in a later part of the article, its main contribution is to understand the effects of time-dependent control actions in flattening the COVID-19 curve. As a result, it is of a higher interest to solve the inverse problem of the model.

This article is organised as follows: "Mathematical Modelling of COVID-19" introduces the mathematical modelling of COVID-19 using SEIRD with feedback for control actions; "Estimation of Reproduction Number and Other Model Parameters" presents the TRR least-squares algorithm used to estimate the reproduction number and other parameters of the model; "Case Study: Modelling the COVID-19 Outbreak in Kazakhstan" provides a case study for the algorithm based on the data in Kazakhstan along with other simulation settings with an extensive discussion of the simulation results; and "Conclusion" concludes the article.

## MATHEMATICAL MODELLING OF COVID-19

Firstly, consider the SEIRD model below, which is modified from the SEIRS model in *Ng & Gui (2020)*,

$$\frac{dS(t)}{dt} = \Lambda - \mu S(t) - \beta(t)S(t)I(t), \tag{1}$$

$$\frac{dE(t)}{dt} = \beta(t)S(t)I(t) - (\mu + \alpha)E(t), \tag{2}$$

$$\frac{dI(t)}{dt} = \alpha E(t) - (\mu + \gamma)I(t) - \phi I(t), \tag{3}$$

$$\frac{dR(t)}{dt} = \gamma I(t) - \mu R(t), \tag{4}$$

$$\frac{dD(t)}{dt} = \phi I(t), \tag{5}$$

where $S(t)$, $E(t)$, $I(t)$, $R(t)$, and $D(t)$ are the compartments representing the susceptible, exposed, infectious, recovered, and deaths population, respectively. The overall population $N(t)$ is established to be $N(t) = S(t) + E(t) + I(t) + R(t) + D(t)$. The constants $\Lambda$ and $\mu$ are the birth rate entering the population and death rate due to non-COVID-19-related conditions, respectively. The parameter $\alpha$ is the rate from being exposed to becoming infectious, and $\gamma$ is the recovery rate. As a result, the incubation and recovery periods can then be computed to be $\tau_{inc} = 1/\alpha$ and $\tau_{rec} = 1/\gamma$. The constant $\phi = \delta(d_{old}N_{old} + d_{oth}(1 - N_{old}))$ is used to describe the population from the infectious compartment that could potentially succumb to the disease, resulting in fatality, where $N_{old}$ represents the fraction of elderly population (above 65 years old), whilst $d_{old}$ and $d_{oth}$ are the fatality rates of the elderly and the rest of the population, respectively. The time to death can be computed using $\tau_{death} = 1/\delta$.

The function $\beta(t)$ represents the transmission rate per S-I contact, such that $\beta_0$ is the initial transmission rate at time $t = 0$, that is $\beta(0) = \beta_0$. Also, define the function $\sigma_{eff}(t) \in [0,1]$ to represent the effective efficacy of the intervention measures introduced to control the spread of the virus and to flatten the curve where it is assumed that $\sigma_{eff}(0) = 0$. Therefore, $\beta(t)$ can be expressed using

$$\beta(t) = \begin{cases} \beta_0, & \text{for } t = 0 \\ \beta_0(1 - \sigma_{eff}(t)), & \text{for } t > 0, \end{cases} \tag{6}$$

where $\sigma_{eff}(t) = \sum_{i=1}^{n} \sigma_i(t)$ such that $\sigma_i(t)$ represents the individual control actions introduced on any particular day $i$ of $n$ days since the first cases have been recorded. Therefore, $\sigma_i(t) > 0$ would indicate that a positive control action has been introduced to reduce the spread of the virus. Conversely, $\sigma_i(t) < 0$ would indicate a negative control action (relaxation of intervention measures) has been taken, such as the lifting of lockdowns and other restrictions, which would then cause the transmission rate of the disease to rise again. This condition will be further explored and studied in "Simulation 3: Estimating $\sigma_{eff}(t)$ and Assessment of Current COVID-19 Profile". The value for $\sigma_i(t)$ is assigned such that the effective efficacy of the control actions are bounded, that is $0 \leq \sigma_{eff}(t) \leq 1$. From (6), it can be seen that if $\sigma_{eff}(t) = 1$, then the transmission rate becomes $\beta(t) = 0$. As a result, the disease would cease to further transmit in the society and is successfully eradicated.

Using (6), the initial basic reproduction number $R_0$ can then be formulated, before any control action are taken (see, for example *Ng & Gui (2020)* for the mathematical proof), using

$$R_0 = \frac{\alpha \Lambda \beta_0}{\mu(\mu + \alpha)(\mu + \gamma + \phi)}. \tag{7}$$

The basic reproduction number $R_0$ represents the average number of people that each infected person is spreading the virus to, that is $R_0 > 1$ indicates that each infected person spreads the virus to more than one other person, hence signifying a growing pandemic whilst $R_0 < 1$ indicates that each infected person spreads the virus to less than one other person, bringing the pandemic under control. Therefore, $\sigma_{eff}(t)$ has a direct effect on the

basic reproduction number for time $t > 0$, where the time-dependent function for the effective reproduction number $R_{\text{eff}}(t)$ can be written using

$$R_{\text{eff}}(t) = \frac{\alpha\Lambda\beta_0(1 - \sigma_{\text{eff}}(t))}{\mu(\mu + \alpha))(\mu + \gamma + \phi)}, \tag{8}$$

where $R_{\text{eff}}(0) = R_0$.

Assuming a closed population with negligible birth and death rates, that is $\Lambda/\mu \approx 1$, $\Lambda \approx 0$, and $\mu \approx 0$, the system (1)–(5) can be rewritten using

$$\frac{dS(t)}{dt} = -\beta(t)S(t)I(t), \tag{9}$$

$$\frac{dE(t)}{dt} = \beta(t)S(t)I(t) - \alpha E(t), \tag{10}$$

$$\frac{dI(t)}{dt} = \alpha E(t) - \gamma I(t) - \phi I(t), \tag{11}$$

$$\frac{dR(t)}{dt} = \gamma I(t), \tag{12}$$

$$\frac{dD(t)}{dt} = \phi I(t), \tag{13}$$

and that the overall population is invariant such that

$$\frac{dN(t)}{dt} = 0 \ \forall t \geq 0 \rightarrow N(0) = N(\infty), \tag{}$$

As a result, the initial basic reproduction number in (7) can be re-expressed using

$$R_0 = \frac{\beta_0}{\gamma + \phi}, \tag{14}$$

and subsequently, the time-dependent effective reproduction number in (8) can be written using

$$R_{\text{eff}}(t) = \frac{\beta_0(1 - \sigma_{\text{eff}}(t))}{\gamma + \phi}. \tag{15}$$

This assumption and model setup will be used for the case study in "Case Study: Modelling the COVID-19 Outbreak in Kazakhstan". Figure 2 shows the block diagram of the model.

## ESTIMATION OF REPRODUCTION NUMBER AND OTHER MODEL PARAMETERS

The system (9)–(13) can be described using the continuous-time dynamical system

$$\dot{x}(t) = f(x(t), p), \quad x(0) = x_0, \tag{16}$$

where $x(t) = (S(t), E(t), I(t), R(t), D(t)) \in \mathbb{R}^5$ are the states and $x_0 = (S(0), E(0), I(0), R(0), D(0)) \in \mathbb{R}^5$ are the initial conditions of the states at time $t = 0$.

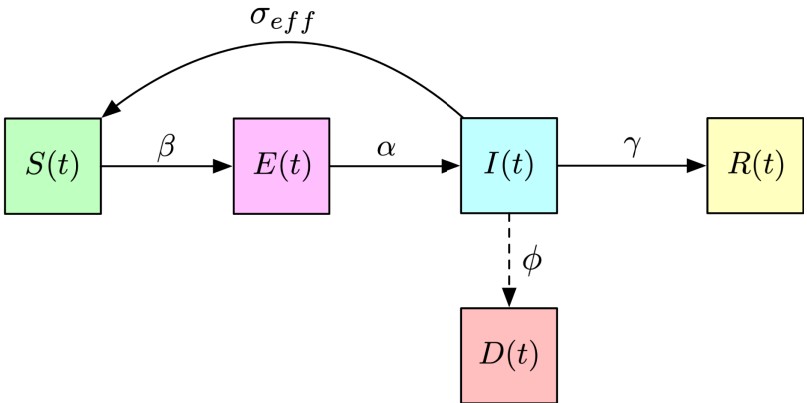

**Figure 2 Block diagram of the SEIRD model used to model the dynamics of COVID-19 in Kazakhstan.**

The vector $p = (\alpha, \beta, \gamma, \delta) \in \mathbb{R}^4$ represents the parameters in the system that are to be estimated. The other parameters, namely $N_{\text{old}}$, $d_{\text{oth}}$, and $d_{\text{old}}$ are assumed to be constants, thus they are omitted from being included into the vector $p$. Therefore, by estimating $\beta$, $\gamma$, and $\delta$, an estimation for $R_{\text{eff}}(t)$ can be produced using (15).

The function $f : U \to \mathbb{R}^5$ is a nonlinear map such that the domain $U$ has the form

$$U = \{(x(t), p) | x_n(t) > 0, p_m > 0\}, \tag{17}$$

for $n = 1, \ldots, 5$ and $m = 1, \ldots, 4$. Equation (17) indicates that all states (or subpopulations in an epidemiological model) are nonnegative given any finite nonnegative initial conditions and that all system parameters are positive. See (*Ng & Gui, 2020*; *Keeling & Rohani, 2008*; *Van den Driessche & Watmough, 2008*) for the proofs on the nonnegativeness, boundedness, and stability of the SEIR model and its variations.

The parameters in the vector $p$ can be estimated over time since the first recorded cases of COVID-19 by solving the following problem in least-squares sense,

$$\min_{p} ||f(x(t), p) - \hat{x}(t)||_2^2 = \min_{p} \sum_{i}^{t} (f(x(i), p) - \hat{x}(i))^2, \tag{18}$$

where $f(x(i),p)$ can be expanded to be

$$f(x(i), p) = \begin{bmatrix} f((S(1), E(1), I(1), R(1), D(1)), p) \\ f((S(2), E(2), I(2), R(2), D(2)), p) \\ \vdots \\ f((S(t), E(t), I(t), R(t), D(t)), p) \end{bmatrix}, \tag{19}$$

and $\hat{x}(t)$ is the predicted or estimated states of the system. It is also established that the parameters are bounded, that is $p_{\min} \le p \le p_{\max}$, to reflect on more realistic real-world values.

## CASE STUDY: MODELLING THE COVID-19 OUTBREAK IN KAZAKHSTAN

At the time of writing, Kazakhstan has passed the first wave of the COVID-19 infected curve but it has still yet to encounter a second wave like many other nations. As such,

**Table 1 Timeline of main events related to COVID-19 in Kazakhstan.**

| Date | Event |
| --- | --- |
| March 13, 2020 (Day 1) | The first two infected cases were confirmed |
| March 16, 2020 (Day 4) | Aggressive control measures were implemented including closure of schools, social distancing, strict border control, limitation of shops opening hours, etc.) |
| March 17, 2020 (Day 5) | State of emergency was declared |
| March 19, 2020 (Day 7) | The whole capital city (Nur-Sultan) was isolated from other parts of the country |
| March 27, 2020 (Day 15) | Operation of enterprises and organisations in Nur-Sultan and Almaty were suspended |
| April 21, 2020 (Day 40) | Nur-Sultan and Almaty eased quarantine regulations, reopened manufacturing facilities, construction industry, and some services |
| May 11, 2020 (Day 60) | Kazakhstan to gradually lift quarantine restrictions. End of state of emergency |
| May 29, 2020 (Day 78) | Checkpoints between cities were removed |
| June 18, 2020 (Day 99) | Checkpoints are being rolled out in districts in North Kazakhstan |
| June 19, 2020 (Day 100) | Quarantine measures are applied for weekends |
| June 22, 2020 (Day 103) | Nur-Sultan shut down all kindergartens |

the data obtained in the country will be an interesting case study for modelling the outbreak of the virus and the prediction of the reproduction number such that its transmission dynamics and the effectiveness of the national-level control actions can be better understood. These results could then be potentially used to predict the future dynamics of the pandemic in Kazakhstan and the suitable control actions to be taken to help flatten the curve. Table 1 shows the major events and control actions taken by the Kazakhstan government up till Day 103 (Jun 22, 2020) in controlling the spread of COVID-19.

## Population facts and initial assumptions of the model

The current population $N$ in Kazakhstan is approximately 18.8 million according to the *United Nations, Department of Economic & Social Affairs, Population Division (2019)* and the fraction of elderly population (65 years of age and above) $N_{old}$ is 8% according to the *Organisation for Economic Co-operation & Development (2018)*. The incubation period $\tau_{inc}$ is set to 5.1 days in line with the report in *Lauer et al. (2020)* and the recovery period $\tau_{rec}$ is 18.8 days according to *Flaxman et al. (2020)*. For this simulation, it is assumed initially that the time to death $\tau_{death}$ is the same as the recovery period, that is $\tau_{death} = \tau_{rec}$, where the patient spends the same amount of time hospitalised whether or not they recover from the disease. Based on the data published by *World Health Organization (2020)*, the fatality rates of the elderly population and nonelderly population are approximated to be 3% and 1.5%, respectively. The initial infectious cases are set to $I(0) = 2$ and it is assumed that $E(0) = 20 \times I(0)$. This initial condition for $E(0)$ is made assuming that 20 persons are exposed to each of the initially infectious person through various physical means and contacts. It is also found through simulations that these assumptions also fit the model well to the initial dynamics of the reported cases. The initial basic

**Table 2 Assumptions of states and parameters used for initial fit of the model.**

| Parameter | Value |
|---|---|
| Overall population, $N(t)$ | $18.8 \times 10^6$ |
| Initial infectious cases, $I(0)$ | 2 |
| Initial exposed cases, $E(0)$ | 40 |
| Initial recovered cases, $R(0)$ | 0 |
| Initial death cases, $D(0)$ | 0 |
| Initial susceptible cases, $S(0)$ | $N(0) - E(0) - I(0)$ |
| Fraction of elderly population, $N_{old}$ | 0.08 |
| Fatality rate of elderly population, $d_{old}$ | 0.03 |
| Fatality rate of nonelderly population, $d_{oth}$ | 0.015 |
| Incubation period, $\tau_{inc}$ | 5.1 days |
| Recovery period, $\tau_{rec}$ | 18.8 days |
| Time to death, $\tau_{death}$ | 18.8 days |
| Initial basic reproduction number, $R_0$ | 3.0 |

reproduction number $R_0$ is assumed to be 3.0. Table 2 shows the summary of the initial assumptions of the states and parameters used to fit the initial trajectory of COVID-19 in Kazakhstan.

However, it is to note that given the limited data available on COVID-19 where most countries only report on the cumulative infectious and deaths cases, only a subset of the states $x(t)$ can be used to estimate the parameters. As such, $w(t) = (I(t),D(t))$ is defined to be used for prediction by the algorithm in (18), which can now be updated and written using

$$\min_p ||f(w(t),p) - \hat{w}(t)||_2^2 = \min_p \sum_i^t (f(w(i),p) - \hat{w}(i))^2, \qquad (20)$$

where $\hat{w}(t)$ represents the predicted variables and $f(w(i),p)$ can be expanded using the similar structure as (19).

The simulations that follow consider the data recorded in Kazakhstan from March 13, 2020 where the first cases were recorded till Day 108 (June 28, 2020), as denoted by the shaded regions in Fig. 1. In "Simulation 1: Estimating Model Parameters Using TRR Assuming Bounded Constraints for β and Constant α, γ, δ" and "Simulation 2: Estimating Model Parameters Using TRR Assuming Bounded Constraints for All Parameters", the model is fitted and its parameters estimated using a time step of 7 days, assuming constant and bounded constraints for time-related parameters, respectively. In "Simulation 3: Estimating $\sigma_{eff}(t)$ and Assessment of Current COVID-19 Profile", the effects of the control actions taken by estimating the value of the control actions efficacy $\sigma_i(t)$ and subsequently, $\sigma_{eff}(t)$, over time in relation to the timeline of COVID-19-related events in the country are analysed. This article will also provide some analysis on the trajectories past Day 108 of the virus for different times of which control actions can be reinstated.

**Table 3 Settings for parameters in Simulation 1.**

| Parameter | Lower bound | Upper bound |
|---|---|---|
| Incubation period, $\tau_{inc} = 1/\alpha$ | 5.1 days | 5.1 days |
| Recovery period, $\tau_{rec} = 1/\gamma$ | 18.8 days | 18.8 days |
| Time to death, $\tau_{death} = 1/\delta$ | 18.8 days | 18.8 days |
| Transmission rate, $\beta(t)$ | 0.01 | 1.00 |

## Simulation 1: estimating model parameters using TRR assuming bounded constraints for β and constant α, γ, δ

First, the fitting of the model is made using the algorithm in (20) assuming that the parameters α, γ, and δ remain constant as set out in Table 2. As for $\beta(t)$, it is assumed to be bounded such that $0.01 \leq \beta(t) \leq 1$. See Table 3.

Figure 3 shows the results of the fitting of the model compared with the data. Figures 3A and 3B show the key plots of the 7-day moving average of the active confirmed cases and the cumulative deaths, respectively. Figure 3C shows the trajectories of all compartments in the SEIRD model. It can be seen that although the active confirmed cases can be predicted relatively well, the predictions for the cumulative deaths are not able to follow the actual data. This is due to the less flexibility in the fitting of the model as all time-related parameters are assumed to be unchanged. Figures 3D–3F show the extended trajectories of Figs. 3A–3C until the model reaches its equilibrium after about 1,000 days since the first cases were reported, assuming that no further control action is taken after Day 108. These results show that the active confirmed cases could peak around Day 500 with approximately 0.6 million cases whilst the cumulative deaths could reach a total of about 50,000. Given that the fitting for the cumulative deaths is underestimated, the actual numbers could potentially rise much higher than the result shown in Fig. 3E. Table 4 shows the estimated reproduction number $R_{eff}(t)$ over time.

## Simulation 2: estimating model parameters using TRR assuming bounded constraints for all parameters

The simulation is then repeated assuming now that all parameters to be estimated are bounded. The incubation period is assumed to be bounded with a range from 2 to 14 days (Lauer et al., 2020). The recovery period and time to death are assumed to be bounded with a range from 1 to 60 days (Flaxman et al., 2020). See Table 5. The results shown in Figs. 4A and 4B depict that both the active confirmed cases and cumulative deaths are able to fit to the actual data much better compared to Figs. 3A and 3B from Simulation 1 in "Simulation 1: Estimating Model Parameters Using TRR Assuming Bounded Constraints for β and Constant α, γ, δ". Figure 4C shows the trajectories of all compartments in the SEIRD model. Table 6 shows the results for the estimated parameters from the optimisation process, which indicate that with the bounded constraints now applied to the time-related parameters, the estimated $R_{eff}(t)$ have more realistic values and they reflect better to the progress of the transmission dynamics of the virus in Kazakhstan. The higher $R_{eff}(t)$ values for time windows starting Days 15 and 22 (highlighted using bold

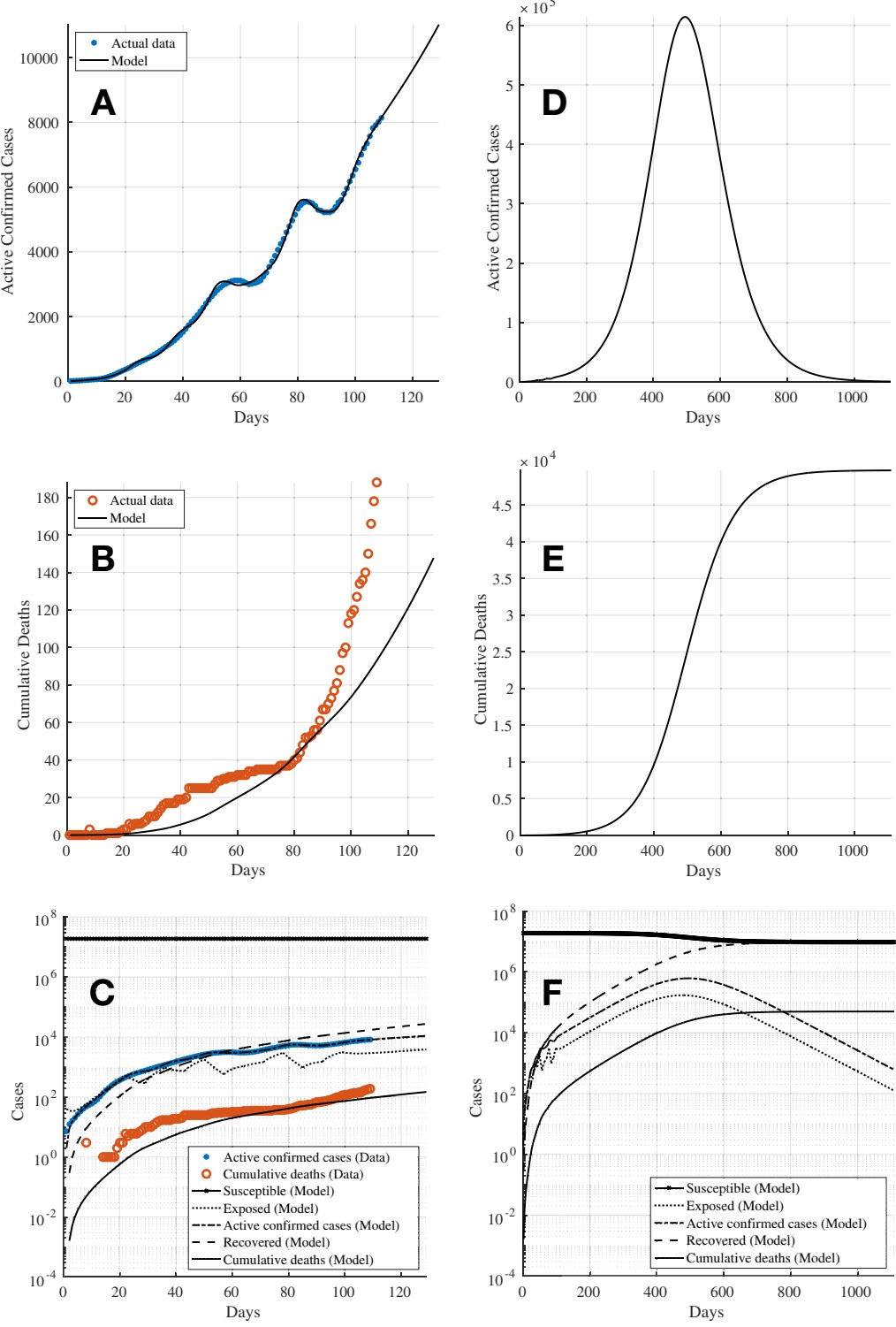

**Figure 3** Simulation 1: The fitting of the model for (A) 7-day moving average active confirmed cases, and (B) cumulative deaths, using bounded transmission rate β and constant incubation, recovery, and time to death periods ($\tau_{inc}$, $\tau_{rec}$, $\tau_{death}$). (C) Shows the trajectories for all compartments in the SEIRD model and (D)–(F) show the extended plots of (A)–(C) until the model achieves equilibrium.

**Table 4 Optimisation results from Simulation 1 using bounded transmission rate and constant time-related parameters**

| Day | Reproduction Number, $R_{eff}(t)$ |
|---|---|
| Initial assumption taken from Table 2 | |
| 1 | $R_0 = 3.00$ |
| 8 | 8.16 |
| 15 | 7.49 |
| 22 | 6.95 |
| 29 | 1.20 |
| 36 | 4.36 |
| 43 | 1.55 |
| 50 | 3.47 |
| 57 | 0.19 |
| 64 | 1.51 |
| 71 | 1.73 |
| 78 | 2.91 |
| 85 | 0.19 |
| 92 | 1.18 |
| 99 | 2.32 |
| 106 | 1.38 |

**Table 5 Settings for parameters in Simulation 2.**

| Parameter | Lower bound | Upper bound |
|---|---|---|
| Incubation period, $\tau_{inc} = 1/\alpha$ | 2 days | 14 days |
| Recovery period, $\tau_{rec} = 1/\gamma$ | 1 day | 60 days |
| Time to death, $\tau_{death} = 1/\delta$ | 1 day | 60 days |
| Transmission rate, $\beta(t)$ | 0.01 | 1.00 |

text in Table 6) could be attributed to potentially lack of testing and record of cases as the country was still coming to grips with the presence of the virus in the society during the earlier stages of the pandemic. Figures 4D–4F show the extended trajectories of Figs. 4A–4C until the model reaches its equilibrium after about 1,200 days since the first cases were reported, assuming that no further control action is taken after Day 108. The results show that with bounded constraints applied to all parameters, the active confirmed cases could peak at slightly over 0.75 million cases around Day 600 whilst the cumulative deaths could reach a total of slightly over 250,000 cases. Nonetheless, both simulations in "Simulation 1: Estimating Model Parameters Using TRR Assuming Bounded Constraints for β and Constant α, γ, δ" and "Simulation 2: Estimating Model Parameters Using TRR Assuming Bounded Constraints for All Parameters" agree that the virus would continue to spread in the society with a mean reproduction number of $R_{eff}(t) \approx 1.44$ for the time window beginning Day 106. Hence, it is essential that effective

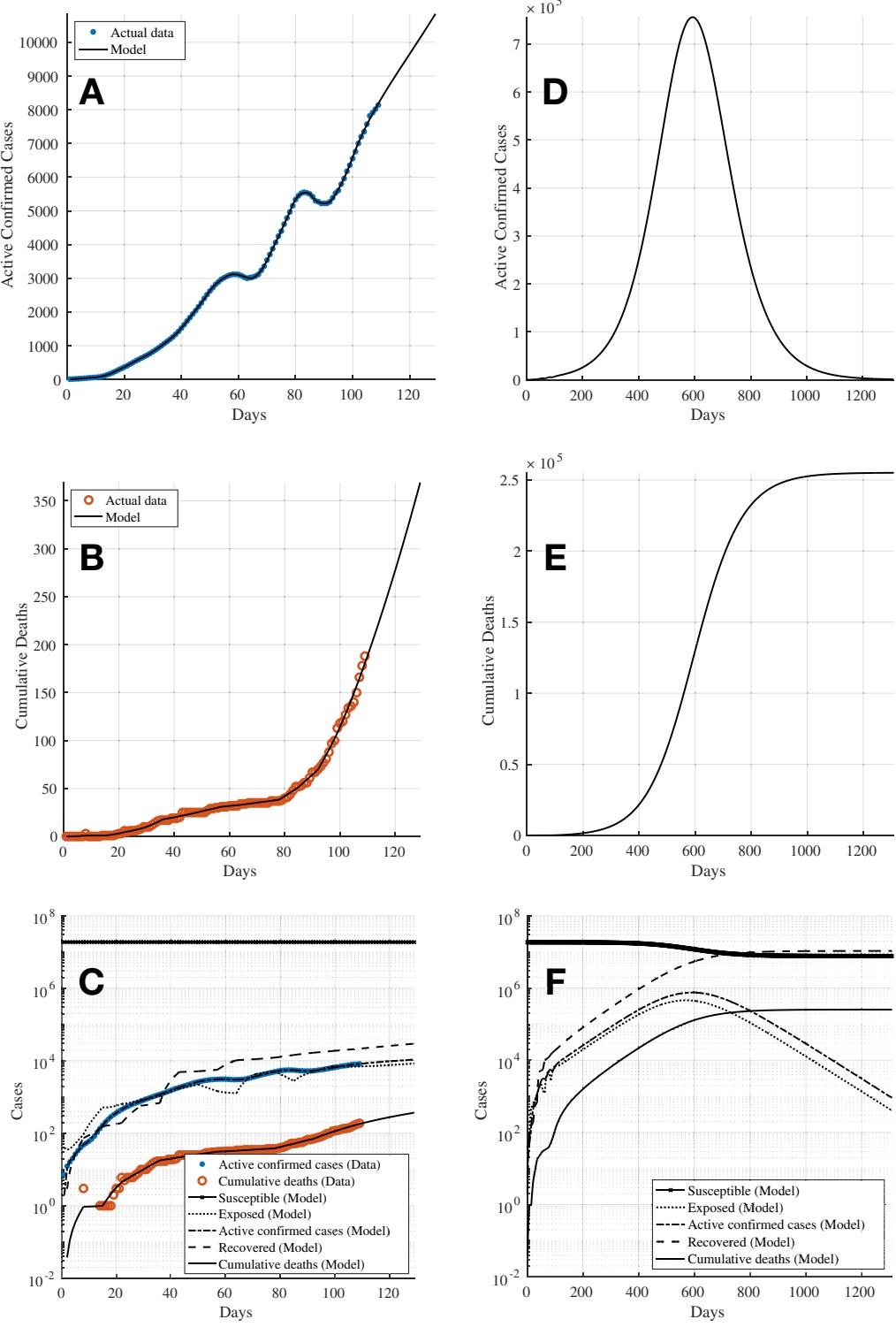

**Figure 4** Simulation 2: The fitting of the model for (A) 7-day moving average active infectious cases, and (B) cumulative deaths, using bounded constraints for transmission rate β and incubation, recovery, and time to death periods ($\tau_{inc}$, $\tau_{rec}$, $\tau_{death}$). (C) Shows the trajectories for all compartments in the SEIRD model and (D)–(F) show the extended plots of (A)–(C) until the model achieves equilibrium.

**Table 6 Optimisation results from Simulation 2 using bounded constraints for all parameters.** Bold values indicate $R_{eff}(t)$ values for time windows starting Days 15 and 22.

| Day | Reproduction number, $R_{eff}(t)$ | Incubation period, $\tau_{inc}$ | Recovery period, $\tau_{rec}$ | Time to death, $\tau_{death}$ |
|---|---|---|---|---|
| Initial values taken from Table 2 | | | | |
| 1 | $R_0 = 3.00$ | 5.10 | 18.80 | 18.80 |
| 8 | 2.33 | 3.06 | 2.36 | 1.00 |
| 15 | **7.48** | 9.39 | 7.48 | 59.71 |
| 22 | **11.71** | 11.82 | 60.00 | 3.29 |
| 29 | 2.29 | 7.16 | 11.21 | 5.05 |
| 36 | 6.52 | 12.04 | 60.00 | 4.15 |
| 43 | 1.31 | 2.00 | 2.49 | 14.41 |
| 50 | 5.92 | 12.89 | 60.00 | 18.45 |
| 57 | 0.40 | 12.91 | 40.37 | 23.81 |
| 64 | 0.94 | 2.00 | 4.52 | 59.11 |
| 71 | 6.29 | 14.00 | 31.73 | 44.18 |
| 78 | 2.92 | 14.00 | 27.66 | 60.00 |
| 85 | 0.17 | 9.64 | 16.93 | 15.76 |
| 92 | 1.85 | 14.00 | 16.74 | 11.01 |
| 99 | 2.74 | 14.00 | 21.20 | 5.67 |
| 106 | 1.51 | 14.00 | 23.92 | 5.46 |

intervention measures and control actions have to be taken to bring the pandemic under control, that is to achieve the reproduction number of $R_{eff}(t) < 1$.

## Simulation 3: estimating $\sigma_{eff}(t)$ and assessment of current COVID-19 profile

To understand the effectiveness of the intervention measures taken by the Kazakhstan government to stop the spread of the virus up to Day 108, the model is simulated by inducing ad-hoc control actions $\sigma_i(t)$ into the model in line with the main events shown in Table 1 as well as to fit to the actual data.

The same fitting parameters in Table 2 are used with the exception of $R_0$ and $I(0)$, where initial values of $R_0 = 3.7$ and $I(0) = 5$ are assumed, respectively. The solid lines in Figs. 5A and 5B represent the estimated data based on the fitting parameters; the solid line in Fig. 5A shows the number of 7-day moving average active confirmed cases whereas the solid line in Fig. 5B shows the number of cumulative deaths. The results show that without any control measures, the curves would rise exponentially indicating that the pandemic would continue to grow. For example, the number of active infected cases would reach about 55,000 by Day 140 whilst the number of deaths would exceed 5,000 by Day 180. It is further noted that this trend of rising cases agrees with the results obtained from Simulation 2 in "Simulation 2: Estimating Model Parameters Using TRR Assuming Bounded Constraints for All Parameters". The unshaded rows in Table 7 show the progress of the efficacy of individual control actions $\sigma_i(t)$, the effective efficacy of control

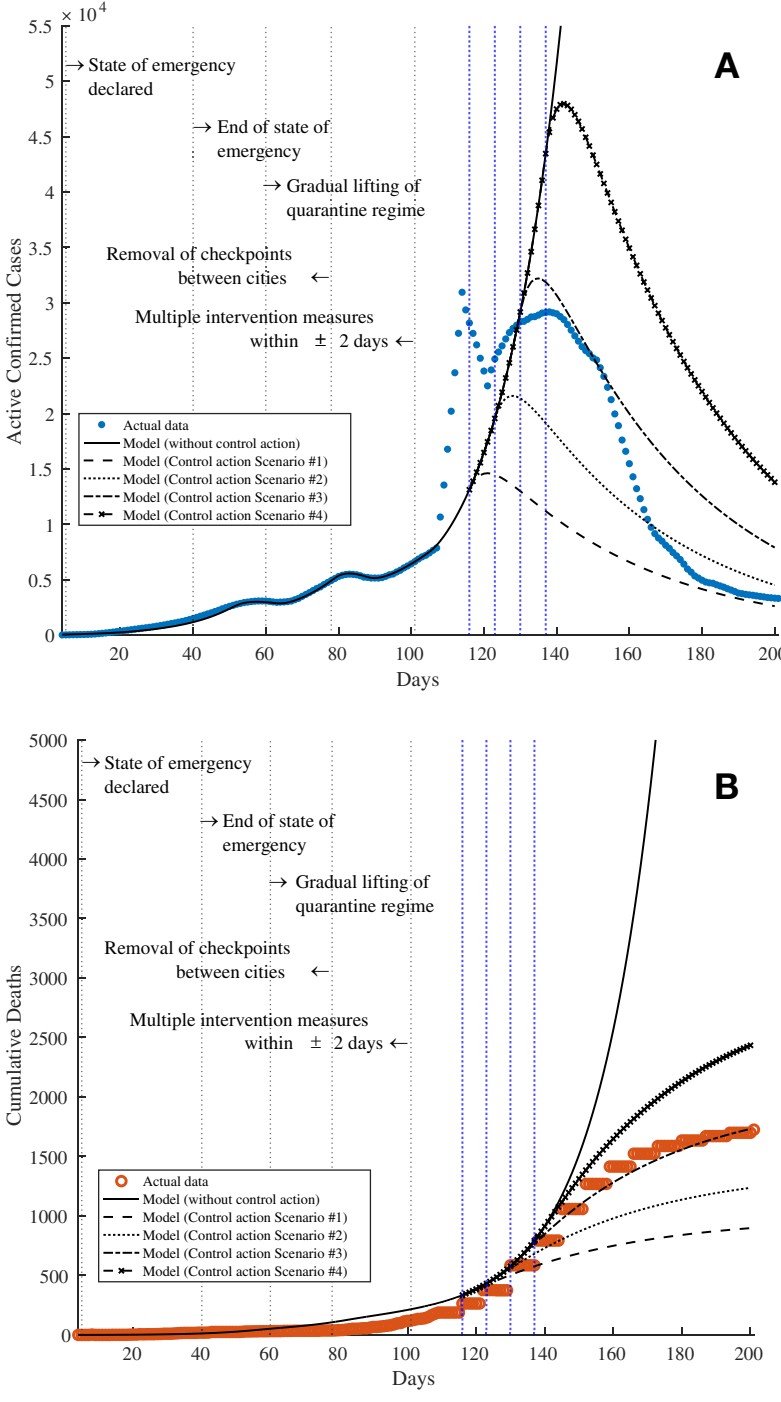

**Figure 5 Simulation 3: The solid lines in both subfigures show the simulation results from fitting the SEIRD model onto the data in Kazakhstan by updating the value of the control action efficacy $\sigma_{eff}(t)$ over time as discussed in "Simulation 3: Estimating $\sigma_{eff}(t)$ and Assessment of Current COVID-19 Profile".** The subfigures show the results for (A) the 7-day moving average active confirmed cases and (B) cumulative deaths, respectively. The dashed, dotted, dashed-dot, and dashed-x lines in both sub-figures show the predictions based on four control action scenarios on Days 116, 123, 130, and 137 (vertical blue dotted lines), respectively, as presented in "Predictions on Reinstating Control and Intervention Measures". The vertical grey dotted lines show the timestamps related to COVID-19 in Kazakhstan as listed in Table 1.

**Table 7 The progress of $R_{eff}(t)$ based on the change in the efficacy of the control actions.**

| Day | Efficacy of Ad-Hoc Control Action, $\sigma_i(t)$ | Effective Efficacy of Control Action, $\sigma_{eff}(t)$ | Reproduction number, $R_{eff}(t)$ |
|---|---|---|---|
| Initial estimation, $R_0$ | 0 | 0 | 3.7 |
| 25 | 0.15 | 0.15 | 3.15 |
| 50 | 0.30 | 0.45 | 2.04 |
| 52 | 0.39 | 0.84 | 0.59 |
| **63** | **−0.48** | **0.36** | **2.37** |
| 79 | 0.54 | 0.9 | 0.36 |
| **88** | **−0.40** | **0.5** | **1.85** |
| **105** | **−0.22** | **0.28** | **2.66** |
| Simulated scenarios of reinstating intervention measures | | | |
| Scenario 1: Day 116 | | | |
| Scenario 2: Day 123 | 0.58 | 0.86 | 0.50 |
| Scenario 3: Day 130 | | | |
| Scenario 4: Day 137 | | | |

actions $\sigma_{eff}(t)$, and the reproduction number $R_{eff}(t)$ over time since the first confirmed cases in the country. The negative values for the efficacy of individual control actions, that is $\sigma_i(t) < 0$ (highlighted using bold text in Table 7) indicate the relaxation of the control actions such as the lifting of lockdowns or state of emergency and the removal of checkpoints between cities. It can also be seen that these coincide with such corresponding events in Table 1, that is Day 40 with the easing of quarantine regulations and the reopening of certain business and industries and 60 with the gradual lifting of quarantines and end of state of emergency ($\sigma_i(t) = -0.48$ on Day 63 in Table 7), and Day 78 with the removal of checkpoints between cities ($\sigma_i(t) = -0.40$ on Day 88 in Table 7), respectively. As a result, a resurgence of the spread of COVID-19 would follow where the $R_{eff}(t)$ increase to 2.37 and 1.85 on Days 63 and 88, respectively. It can be observed that there is a lag of 10–20 days between the announcements of the enforcements or lifting of control actions to the actual effects being recorded via the confirmed cases. This could be due to a few reasons, namely (i) the incubation time with a median of 5.1 days before a patient starts to show symptoms and become infectious; (ii) the time delay incurred for the population and the industries involved to respond and act according to the control actions announcements, which could take one to 2 weeks.

## Comparison of simulation results

The simulation results obtained in "Simulation 1: Estimating Model Parameters Using TRR Assuming Bounded Constraints for $\beta$ and Constant $\alpha$, $\gamma$, $\delta$" and "Simulation 2: Estimating Model Parameters Using TRR Assuming Bounded Constraints for All Parameters", and "Simulation 3: Estimating $\sigma_{eff}(t)$ and Assessment of Current COVID-19 Profile" are then compared to see how well each of the simulation methods are able to fit to the actual data. See Fig. 6, where Figs. 6A and 6B compare the model fitting error for the active confirmed cases and cumulative deaths of the three simulation methods, respectively.

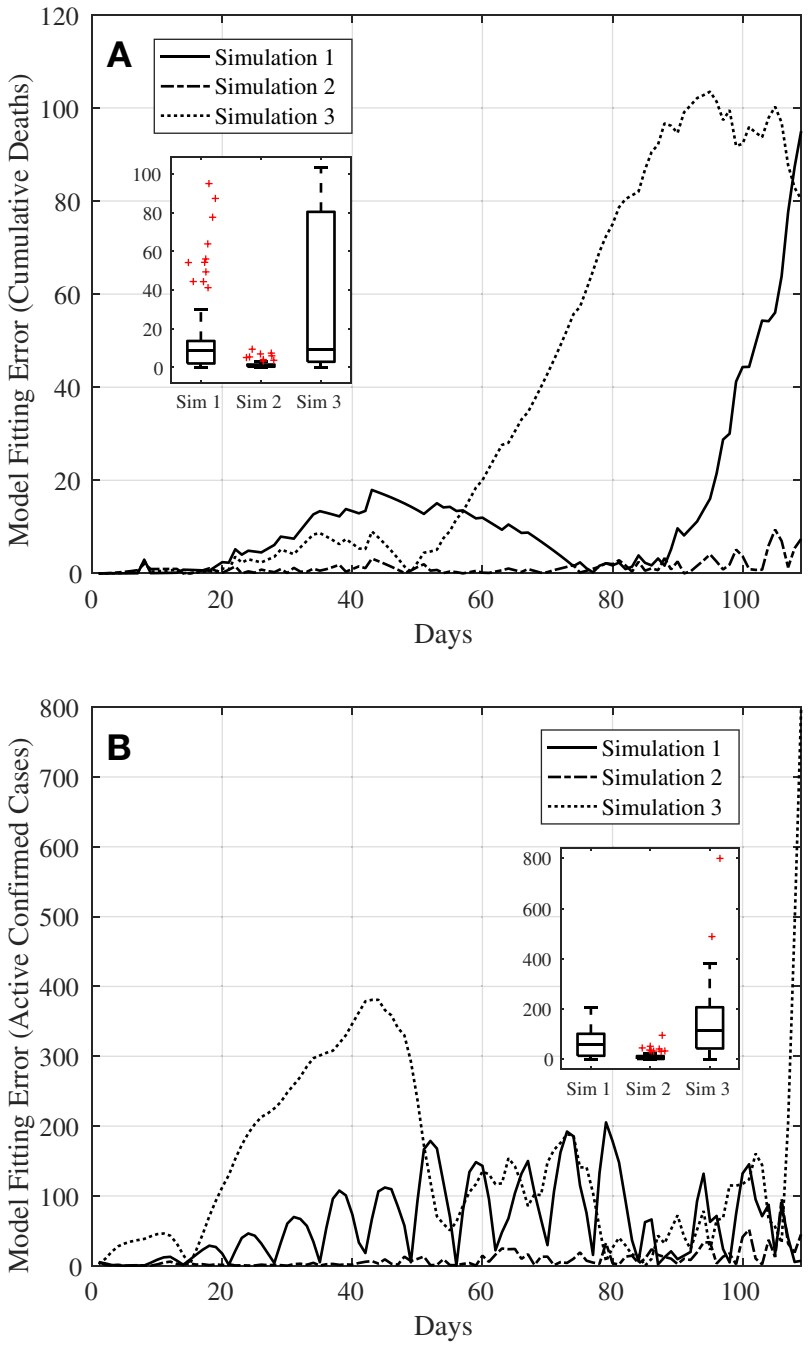

**Figure 6 Plots comparing the simulation results from the three simulation methods in Simulations 1, 2, and 3, respectively.** (A) Shows the model fitting error for active confirmed cases whilst (B) shows the model fitting error for cumulative deaths. The inset boxplot in each subfigure shows the statistical analysis of the three fitting methods.                

For the model fitting error for the active confirmed cases, Fig. 6A shows that all three methods manage to achieve errors with a median of less than 150 cases; medians of 57.13, 3.67, and 115.10 for Simulations 1, 2, and 3, respectively. Whilst Simulation 2 produces the smallest error amongst the three simulation methods, this could also indicate that the

model is overfitted. This might not be able to inform well on the effectiveness of the control action taken on the effective reproduction number $R_{\text{eff}}(t)$ in relation to the timeline of the main events tabulated in Table 1.

As for the model fitting error for the cumulative deaths, Fig. 6B shows that again all three methods manage to achieve errors with relatively low median of less than ten cases (medians of 8.70, 0.82, and 9.10 for Simulations 1, 2, and 3, respectively) even though Simulation 3 produces a much larger interquartile range (IQR) compared to Simulations 1 and 2. And as with the model fitting error for the active confirmed cases, the model obtained from Simulation 2 could be overfitted.

Despite results from Simulation 3 producing slightly larger errors with a higher median and larger IQR compared to Simulations 1 and 2, this method will be used in "Predictions on Reinstating Control and Intervention Measures" to simulate and predict the future trajectories of the dynamics from reinstating control and intervention measures to bring the pandemic under control. The main reason for this is that the method used in Simulation 3 allows for manual setting of ad-hoc control actions in the model in line with the timeline of the main events in Table 1. Hence, this will help to inform and allow us to understand more on the effectiveness of time-dependent control actions taken in reducing the value of $R_{\text{eff}}(t)$, hence bringing the outbreak of COVID-19 under control.

## Predictions on reinstating control and intervention measures

Assume now that following the results obtained via Simulation 3 in "Simulation 3: Estimating $\sigma_{\text{eff}}(t)$ and Assessment of Current COVID-19 Profile", the reproduction number is to be reduced to $R_{\text{eff}}(t) < 1$ such that the spread of the virus is under control. As a result, the necessary intervention measures have to be reinstated. Hence, four scenarios are simulated where the intervention measures would be reinstated on Days 116, 123, 130, and 137, respectively by setting $\sigma_i(t) = 0.58$ such that the reproduction number becomes $R_{\text{eff}}(t) = 0.50$ as shown using the shaded rows in Table 7.

With Scenario 1, the number of active confirmed cases would reach its peak around Day 120 with 14,500 cases before they gradually reduce indicating that the pandemic is under control. For Scenarios 2–4, the active confirmed cases peak on Days 128, 135, and 142, with about 21,600, 32,200, and 48,000 cases, respectively. These data are shown using the dashed, dotted, dashed-dot, and dashed-x lines in Figs. 5A and 7A, where the latter shows the trajectories until the model reaches equilibrium. Similarly, the dashed, dotted, dashed-dot, and dashed-x lines in Figs. 5B and 7B show the total number of deaths for these four scenarios. The total number of deaths are estimated to be approximately 1,000, 1,400, 2,000, and 2,900 for Scenarios 1–4, respectively. On a further note, although the active confirmed cases would reach the equilibrium around Day 400 with almost the same value of approximately 45–130 cases (see Fig. 7A), the simulation results from the four scenarios clearly show that with every delay of 7 days in reinstating the intervention measures, the peaks of the active confirmed cases and cumulative deaths (see Fig. 7B) would increase exponentially. Therefore, it is obvious that the sooner the reinstating of intervention measures are implemented, the better the outcomes of the situation, especially to reduce deaths and save precious lives.

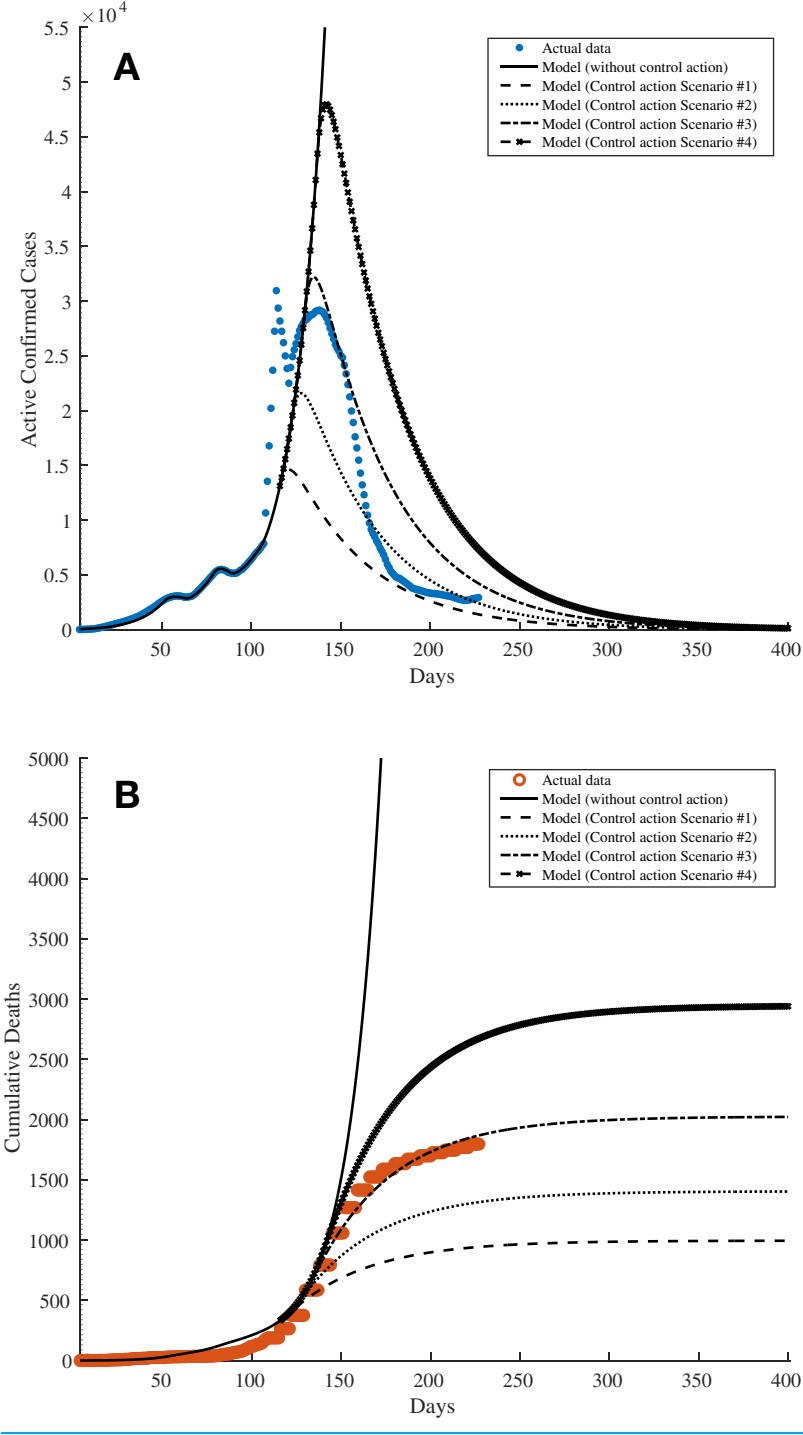

**Figure 7 Simulation results showing the predictions based on four control action scenarios until the model reaches an equilibrium.** The subfigures show the predictions for (A) the 7-day moving average active confirmed cases and (B) cumulative deaths, respectively. The dashed, dotted, dashed-dot, and dashed-x lines in both subfigures show the predictions based on four control action scenarios on Days 116, 123, 130, and 137, respectively, as presented in "Predictions on Reinstating Control and Intervention Measures".

From these simulations, it is evident that the current situation in Kazakhstan is developing quite closely to Scenario 3 for both active confirmed and cumulative deaths cases. As a result, perhaps these information can be potentially used to address and control the pandemic during an unfortunate event of a second wave. In addition, the results shown in Table 7 can also be further analysed to model the mobility and dynamics of the population in the country and its various districts in responding to various levels of intervention measures taken by the authorities.

## CONCLUSION

This article has discussed and presented a methodology to assess the current COVID-19 pandemic profile and to use those information to provide critical information on efficacy and effects of time-dependent control actions in flattening the curve of COVID-19. A modified SEIRD model was used in this study and the parameters of the mathematical model as well as the reproduction number were estimated, for both constant and bounded constraints conditions for time-related parameters, using the trust-region-reflective (TTR) algorithm where the data in Kazakhstan were used as a case study. A further analysis was carried out by inducing ad-hoc control actions into the model and to determine how well they correspond to actual events and control actions recorded in the country. Four scenarios were further simulated to provide understanding about the effects of reinstating intervention measures taken 7 days apart of each other onto the active confirmed and cumulative deaths cases. The results show that any delay in reinstating the intervention measures would increase the peak of the active confirmed cases and also the cumulative deaths exponentially. Of course, the quantitative analysis in this article is highly dependent on the accuracy of the input data. With the limited data available and using the presented modelling, assessment, and prediction techniques, it is hope that this research is able to inform the transmission dynamics of the virus and provide some useful information and analyses for the COVID-19 situation in Kazakhstan.

### Funding
The authors received no funding for this work.

### Competing Interests
Kok Yew Ng is an Academic Editor for PeerJ.

### Author Contributions

- Ton Duc Do conceived and designed the experiments, performed the experiments, analyzed the data, authored or reviewed drafts of the paper, and approved the final draft.
- Meei Mei Gui analyzed the data, prepared figures and/or tables, authored or reviewed drafts of the paper, and approved the final draft.
- Kok Yew Ng conceived and designed the experiments, performed the experiments, analyzed the data, prepared figures and/or tables, authored or reviewed drafts of the paper, and approved the final draft.
## Data Availability
The MATLAB/Simulink files are available in the Supplemental Files.

## Supplemental Information
Supplemental information for this article can be found online at http://dx.doi.org/10.7717/peerj.10806#supplemental-information.

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
