# Peer review of "Assessing the effects of time-dependent restrictions and control actions to flatten the curve of COVID-19 in Kazakhstan"

_PeerJ, doi:10.7717/peerj.10806_

## Round 0.1 · original submission · Major Revisions

Dear authors,

Based on the reviewers' comments, I think your paper has high standards to be published in PeerJ, once some issues are solved by you. Please, see the comments below so as to have more information.

Best regards,
Dr Palazón-Bru

Reviewer 1 ·

Basic reporting

The manuscript is well-written, with extensive literature review. Figures and tables are sufficient to illustrate the results of this study.

Experimental design

No comment

Validity of the findings

No comment

Additional comments

There are many similar existing studies on the topics and the current study fails to add novel insights. Following are three minor comments:
(1) Normally, \Delta is not used as a model coefficient;
(2) It would be better to explicitly indicate the dates in x-axis of Figure 1
(3) The values of incubation periods, recovery periods and time to death those values lie outside the confidence intervals predicted by most studies, which are not plausible.

Reviewer 2 ·

Basic reporting

no comment

Experimental design

no comment

Validity of the findings

no comment

Additional comments

In this manuscript, the authors present a SEIRD model with both time-invariant and time-varying transmission rate for the control and assessment of time-dependent restrictions put in place by governments during the COVID-19 pandemic.
The paper is well structured; the presentation of the topics is clear, and a very professional English language is used. The problem is clearly explained in the introduction where compartmental models SIR and SEIR are reported in the modelling of COVID-19 pandemics. The same, however, does not happen for the SEIRD model with constant and time-dependent parameters. Although it is difficult to follow the significant number of preprints and papers appearing about this subject, a few references are needed. SEIRD has been used in COVID-19 pandemic both with constant parameters (for example https://doi.org/10.1101/2020.05.10.20083683, https://doi.org/10.3346/jkms.2020.35.e317) as well as time-dependent parameters, see https://doi.org/10.1371/journal.pone.0237417 and references therein.
Moreover, while the choice of the Trust-Region Reflective method for the solution of the inverse problem (parameter estimation) is well-motivated, nothing appears, in the introduction, concerning the solution of the forward problem (ODE). A discussion about this point would improve the whole introduction.
Concerning the Mathematical modelling (section 2) the SEIRD model and reproduction number are presented for the most general case and then studied for a close population. It is advisable to state the SEIRD for a close population and carefully check the correspondence of equations (9)-(13) with the modelling simulation reported in table 6 (further discussion in the next paragraphs).
The paper presents a case study based on Kazakhstan epidemic data. The experimental section is well designed, and the Initial assumptions are reported in detail. However, the initial value of the exposed E(0)=20 I(0) appears to be entirely arbitrary. The motivation and the influence of this choice on the final results need further discussion.
The description of the parameter estimation procedure and simulation 1 (4.2) are clear and accurate. On the contrary paragraphs, 4.3 and 4.4 need improvement.
The plots in figure 4(a-c) and the data in table 6 are obtained by splitting the simulation into 14 days intervals. Since the parameters \alpha-\beta-\gamma-\delta are constant on each interval and change from one interval to another, they are piecewise constant functions throughout the time domain. The text does not evidence this fact, which should be highlighted by reporting in a table the values of the parameters computed at each time interval.
Moreover, from the text, it is not clear whether the control actions sigma(t) are assigned/guessed or obtained through a parameter estimation procedure. This aspect should be clarified in detail by explaining how the numbers in the second column of Table 7 are obtained. Moreover, if such values are assigned, pèlease explain the reasoning behind these choices.
Table 7 is not easy to read, especially in the shaded part, the link to the four scenarios described in the text could help the reader.

Finally, the included Matlab code that computes data in Table 6 and plots Figure 4 (a),(b),(c), performs the optimization step using only Infected and Dead populations. (see runNonlinear.m at line 32)
Please amend this by correcting the code and the table/figures or writing a consistent expression in equation 20, explaining why the Recovered data should be removed.


Typos:
Ln 88: /tiem/. /time/
Ln 248: /apparent/. /evident/

Decision

The present manuscript tackles an actual crucial topic, and even though plenty of literature is available with the same purpose, the methodology presents a novelty aspect. The paper is well written, and it has a good structure. The results are engaging, but there are essential aspects in the reference, mathematical model and experimental section that need to be clarified before it can be accepted for publication.
Accept after revision.

Reviewer 3 ·

Basic reporting

“Figure 2. Block diagram of the SEIRD model used to model the dynamics of COVID-19 in Kazakhstan” is not present.

Line 23: ….."to better plan for and control of the subsequent waves of the pandemic"….
Seems “of” is extra here.

Line 26: …"which is one of the critical measurements"… measures may be better.

Line 83-85: The sentence needs to be reworded to make it clear.

Line 88: ----"tiem-related"--- must be time-related

Line 105: ----"parameters"-----change it to "parameter"

Give reference for equation (7), it can not be arrived using equation (6)

Line 130-134: Split the sentence into smaller sentences

Line 147: for fatality rates provide reference

Line 155: -----"estimates"---change it to estimate

Use passive for writing paper and avoid we, I etc.

Experimental design

For equation (14) one more assumption needs to be mentioned that is birth and death rates are almost equal in addition to being negligible (as has been mentioned).

In equation (6), the intervention parameter must always be positive and up to 1, else the transmission rate can lead to higher values in partial intervention case than in no intervention case.

Also consider adding statistical analysis to see the variation between model results and reported data.

Validity of the findings

Results in Line 213-219 difficult to justify as intervention parameter is showing negative vales.

Without statistical analysis difficult to comment on validity of results.

Additional comments

Please revise the work as per the mentioned comments.
Thanks!!!

---

## Round 0.2 · Minor Revisions

Still pending some minor changes suggested by one of the reviewers.

Reviewer 2 ·

Basic reporting

no comment

Experimental design

no comment

Validity of the findings

no comment

Additional comments

The authors have correctly included all the suggested amendments. The paper is significantly improved.

Reviewer 3 ·

Basic reporting

All comments addressed

Experimental design

The revised σ(t) ∈ (−∞,1] is difficult to justify especially on the lower bound. As it will make the modified beta reaching to infinity. And such large values of beta will infect the global population within a day, which is not the characteristic of at least this COVID-19.

Moreover, the removal of control interventions should bring the transmission to natural value of beta, meaning σ should become zero. Making σ negative means that the removal of control interventions are making transmission rates increase beyond the non-intervention level.

It would be good to provide references for effective beta values going beyond non-intervention values while removing control interventions and justification for this scenario.

Validity of the findings

Depend on the validity and justification of the σ(t) ∈ (−∞,1]

Additional comments

Dear Authors,

Please address the comments.

---

## Round 0.3 · accepted · Accept

All the reviewers' comments have been correctly addressed.

Reviewer 3 ·

Basic reporting

All comments addressed

Experimental design

All comments addressed

Validity of the findings

All comments addressed

Additional comments

Dear Authors,

Thanks for your efforts